# An Empirical Evaluation of NVM-Aware File Systems on Intel Optane DC Persistent Memory Modules †

**Guangyu Zhu [1,2]** , **Jaehyun Han [1]**, **Sangjin Lee [1] and Yongseok Son [1,2,*]**

1   Department of Computer Science and Engineering, Chung-Ang University, Seoul 06974, Korea;
    zgy29@cau.ac.kr (G.Z.); whffu2421@cau.ac.kr (J.H.); tkdwls0727@cau.ac.kr (S.L.)
2   CAU Institute for Innovative Talent of Big Data, Department of Computer Science and Engineering,
    Chung-Ang University, Seoul 06974, Korea
*   Correspondence: sysganda@cau.ac.kr
†   This paper is an extended version of our paper published in International Conference on Information
    Networking (ICOIN'21) .

**Abstract:** The emergence of non-volatile memories (NVM) brings new opportunities and challenges to data management system design. As an important part of the data management systems, several new file systems are developed to take advantage of the characteristics of NVM. However, these NVM-aware file systems are usually designed and evaluated based on simulations or emulations. In order to explore the performance and characteristics of these file systems on real hardware, in this article, we provide an empirical evaluation of NVM-aware file systems on the first commercially available byte-addressable NVM (i.e., the Intel Optane DC Persistent Memory Module (DCPMM)). First, to compare the performance difference between traditional file systems and NVM-aware file systems, we evaluate the performance of Ext4, XFS, F2FS, Ext4-DAX, XFS-DAX, and NOVA file systems on DCPMMs. To compare DCPMMs with other secondary storage devices, we also conduct the same evaluations on Optane SSDs and NAND-flash SSDs. Second, we observe how remote NUMA node access and device mapper striping affect the performance of DCPMMs. Finally, we evaluate the performance of the database (i.e., MySQL) on DCPMMs with Ext4 and Ext4-DAX file systems. We summarize several observations from the evaluation results and performance analysis. We anticipate that these observations will provide implications for various memory and storage systems.

**Keywords:** non-volatile memory; file systems; performance

## 1. Introduction

Emerging non-volatile memory (NVM) technologies, such as spin–torque transfer memory [1], phase change memory [2], resistive memory [3], and Intel and Micron's 3D XPoint technology [4] promise to revolutionize I/O performance. NVM brings persistence at latencies within an order of magnitude of DRAM [5,6] and creates a new level in the storage hierarchy between DRAM and traditional secondary storage devices.

As the emergence of new hardware and platforms always leads to reconsideration of how to design data management systems, the advent of NVM brings new opportunities as well as new challenges to system design. The file system, as an important part of the data management systems, is a hot topic for NVM application research. Many researchers have designed and developed new NVM-aware file systems that make better use of NVM features, such as BPFS [7], PMFS [8], and NOVA [9,10]. However, because the design and evaluation of previous studies were usually based on simulations and emulations, their exact performance in working on real NVM devices is unknown. Recently, the first byte-addressable NVM product, Intel Optane DC Persistent Memory Module (DCPMM) [11], has become commercially available. An evaluation on the real hardware can enable accurate

assessment of the impact of NVM-aware file systems on applications and provide hints for future system designs.

In this article, in order to understand the performance differences between NVM-aware file systems and traditional file systems on NVM devices, we evaluate the performance of NVM-aware file systems and traditional file systems at various workloads on Intel Optane DCPMM. Because new devices do not always guarantee better results in all cases, in order to help researchers and system designers make better choices about which storage devices to use when designing future systems, we also evaluate the performance of these file systems on Intel Optane SSDs [12] and NVMe SSDs to compare the performance differences and characteristics of each file system under different storage devices. From the evaluation results, we obtain the following observations that may be useful for future system design: (1) Direct Access has a significant improvement on file system write performance, but has a smaller impact on read performance. (2) Concurrent write access can degrade the performance of DCPMMs while concurrent read access does not. (3) Access from remote sockets reduces the write performance of DCPMMs but has little impact on the read performance. (4) Redesigning low overhead metadata management strategy of file systems is needed to fully utilize the performance of DCPMMs. (5) Organizing DCPMMs from different sockets as a logical device can also bring out the full performance of the devices. (6) DCPMM is better able to show its strengths for small size requests. (7) Even with the same system APIs, we need to consider their different impact on normal block devices and NVM devices under different hardware platforms.

This paper is an extended version of our previous work [13]. In the previous work, we evaluated and analyzed the performance of file systems on DCPMMs by using file system benchmarks (e.g., FIO [14] and Filebench [15]). In this paper, we evaluate and analyze the performance of the database on DCPMMs with different configurations. In addition, we analyze the results of each evaluation in more detail.

## 2. Background and Related Work

This article focuses on investigating the performance characteristics of traditional and NVM-aware file systems on DCPMMs and comparing DCPMMs with other secondary storage devices, thereby providing assistance in the design of future storage systems. In this section, we first briefly present the features of Intel Optane DCPMM. Next, we explain the similarities and differences between NVM-aware and traditional file systems. Finally, we discuss the related work.

### 2.1. Intel Optane DC Persistent Memory Modules

Intel Optane DCPMM is the first commercially available byte-addressable NVM product that is based on 3D XPoint technology [4]. DCPMMs connect to the CPU via the integrated memory controller (iMC) and DDR-T protocol [16]. The iMC maintains read and write pending queues for each DCPMM. DDR-T uses a 72-bit data bus and exchanges data in a cache line unit (64-byte). As a result, the CPU can skip the page cache and perform memory I/O operations directly on the DCPMM. The basic access unit inside the DCPMM is 256 bytes; thus, any write request less than 256 bytes becomes a read–modify–write operation, increasing the write amplification. To reduce the write amplification caused by the two different access granularities, DCPMMs have a small internal write-combining buffer for merging adjacent writes. To ensure persistence, write pending queues of DCPMM and the internal write buffers are placed into Intel's asynchronous DRAM refresh (ADR) domain, which ensures that CPU stores that reach to the write pending queue will survive a power failure [17].

DCPMM can be used as memory or as persistent storage, depending on the operation mode in which it is set to work [11].

**Memory Mode**: In this mode, DCPMMs and DRAM are fused into a single volatile memory space. Because DRAM outperforms DCPMM, DRAM is used as a direct-mapped write-back cache for DCPMM. It should be noted that, even though the storage medium

of DCPMM is non-volatile, for data security, the data in DCPMMs is intentionally crypto-graphically erased between power cycles in memory mode.

**App Direct Mode**: In this mode, DCPMMs are exposed to the system as permanent storage devices. DCPMMs on the same CPU socket can be grouped into an interleaved group. An interleaved group is similar to a RAID-0 device at the hardware level, which increases bandwidth by data striping. In this article, because we are concerned with exploring how file systems work on DCPMMs, we configure our DCPMMs to the App Direct mode.

Intel has another product that uses 3D XPoint as storage media: the Intel Optane SSD. Unlike DCPMMs the Optane SSD is connected to the machine via PCIe and NVMe interfaces, so it cannot be used as memory. It is more like a NAND-flash SSD, but with lower latency and no garbage collection.

### 2.2. NVM-Aware File Systems

Previous studies have proven that traditional file systems do not provide the full performance of low-latency storage devices [18–20]. In order to better utilize the low latency, high bandwidth, and byte-addressable characteristics of NVM devices, some NVM-aware file systems, such as XFS-DAX [21], Ext4-DAX [22], BPFS [7], PMFS [8], and NOVA [9] were developed.

The NVM-aware file systems usually use a technique called *Direct Access* (DAX) [23] to bypass the operating system page cache and I/O stack, thus reducing the overhead and latency caused by the software. DAX enables applications to map files on a NVM-aware file system into their address space, so applications can access data directly through memory load and store instructions. In contrast, traditional file systems do not support DAX. In addition, some traditional file systems are designed to rely on the sector write atomicity provided by the storage device. To support these file systems, DCPMMs need to maintain an additional software layer called the Block Translation Table (BTT) to guarantee the atomic sector update [24]. The overhead of maintaining BTT further widens the performance gap between traditional file systems and NVM-aware file systems.

XFS-DAX and Ext4-DAX are NVM-aware versions of XFS and Ext4, which support Direct Access. BPFS is a NVM-aware file system that uses a shadow paging technique to reduce write amplification and data copy. PMFS provides consistent and durable updates to file system metadata and enables DAX to applications via the `mmap` interface. NOVA adopts the techniques of log-structured file systems to provide a separate log for each inode, thus delivering high performance while ensuring strong consistency.

### 2.3. Related Work

**NVM-aware file systems.** In order to make the file system better adapted to NVM devices, a number of file systems designed specifically for NVM have emerged. To take advantage of the byte-addressable capability of NVM, BPFS [7] can process fine-grained atomic updates to NVM. Therefore, BPFS provides good performance with strong reliability. PMFS [8] is a light-weight POSIX file system that can enable Direct Access by applications. It provides fine-grained logging for consistency by utilizing the processor's paging and memory ordering features. NOVA [9] aims to maximize performance on hybrid memory systems with strong consistency. It uses the techniques of log-structured file systems to provide a separate log for each file's inode, thus exploiting the fast random access of NVM. NOVA-Fortis [10] is a version of NOVA that provides greater reliability. Compared to NOVA, NOVA-Fortis additionally uses snapshots, checksums, replication, and RAID-4 parity protection. In contrast to implementing new file systems, our article focuses on verifying on real hardware as to whether these NVM-ware file system designs can actually be useful in NVM.

**NVM management.** Because memory controller and CPU may reorder memory writes, we need use memory fence and cache line instructions to guarantee data consistency. NV-Tree [25] is a variant of B+ Tree that was designed for NVM. NV-Tree can reduce

the number of cache line flushes that we need to maintain data consistency; as a result, it can outperform the state-of-the-art consistent tree by up to 12X. AsymNVM [26] is an architecture that helps with sharing NVM devices to multiple servers and provides recoverable persistent data structures. This architecture decouples server machines from NVM storage, enabling increased NVM utilization. Combining DRAM and NVM into a hybrid memory can provide a large main memory capacity. However, the problem arises of how to place the data. RTHMS [27] provides an algorithm for data placement on hybrid memory systems. RTHMS analyzes applications in advance and then helps programmers to determine how to place memory objects. Similarly, Dulloor et al. [28] propose a memory management infrastructure that makes it easy for the programmer to decide where to put the corresponding data when allocating memory. They also provide a profiling tool that automatically analyzes the data access pattern and optimizes the data placement. Our article is in line with these studies in terms of finding best practices for using NVM. In contrast, we focus more on using NVM as persistent storage instead of the main memory.

**Evaluation of fast storage devices.** With the advent of new fast storage devices, researchers have done a lot of work to evaluate the performance of the applications working on new devices. Weiland et al. [29] explore the performance of high-performance scientific applications working on DCPMMs in both Memory and App Direct modes. The evaluation results show that a larger memory capacity has better efficiency for scientific applications. Gotze et al. [30] evaluate several tree-based data structures on DCPMMs. From the result, they summarize 16 insights to help design data structures for NVM devices. Wu et al. [31] conduct a performance evaluation of the Intel Optane SSD using the micro benchmark. They propose seven best practices for using NVM-based block devices by analyzing the evaluation result and the Intel Optane SSDs. The comparison reveals that if these best practices are violated, it will have a significant impact on the throughput and latency. Son et al. [32] perform a thorough performance evaluation of NVMe NAND-flash SSDs and compared them to SATA SSDs. Xu et al. [33] evaluate the performance of commonly used database systems (i.e., MySQL, Cassandra, and MongoDB) on NVMe drivers. They present a detailed analysis of the characteristics of NVMe drivers. The result shows that NVMe-backed databases can outperform SATA-based databases by up to eight times. Our study is in line with these studies in terms of evaluating the performance of emerging fast storage devices. In contrast, our study concentrates on comparing the performance characteristics between traditional block-based file systems and NVM-aware file systems on DCPMMs. In addition, we also compare the DCPMMs with Intel Optane SSDs and NAND-flash SSDs.

## 3. Evaluation Settings and Methodology

In this section, we introduce our experiment platform and the methodologies that we used to conduct the evaluations. The specifications of our experiment platform are shown in Table 1. Our machine has four sockets, and each socket has a CPU with 16 cores. During the evaluation, we disable the hyper-threading machine. We have two Intel Optane DCPMMs, which are installed in two different sockets. In this article, we always configure DCPMM to the App Direct mode because we concentrate on observing the performance of file systems on DCPMMs. In order to observe the performance differences of file systems when working on different storage devices, we also use one Intel Optane SSD and one Intel P4610 NAND-flash SSD, which sit on the PCIe interface.

The traditional file systems that we evaluate in this article are Ext4, XFS, and F2FS [34]. The NVM-aware file systems that we evaluate are Ext4-DAX, XFS-DAX, and NOVA. Ext4 and XFS are two mature file systems designed for block devices that have been developed for over 20 years. F2FS is a file system designed to make better use of modern NAND-flash storage devices. Ext4-DAX, XFS-DAX, and NOVA are NVM-aware file systems that can enable the Direct Access feature.

First, we use FIO [14] as a micro benchmark to evaluate the performance of different file systems on DCPMMs, Optane SSDs, and NAND-flash SSDs. If not specified, we only

use processors on the same socket with the DCPMM to eliminate the influence of NUMA effects. FIO generates four simple workloads, such as sequential read, sequential write, random read, and random write. The I/O engine is set to `sync`. All workloads use a 1 GB file size for each thread, and the request size (i.e., the blocksize option) is fixed to 4 KB. We issue a `fsync()` system call after each request in the case of write workloads. Thus, we avoid the impact of delayed allocation. Each test case runs for 30 s. The number of threads varies from 1 to 16, and each thread accesses a different file.

**Table 1.** Evaluation platform specifications.

| | |
|---|---|
| **Processor** | Intel Xeon Gold 6242 Processor |
| **Cores** | 16 cores $\times$ 4 sockets (hyper-threading disabled) |
| **Memory Controller** | 2 iMCs $\times$ 3 channels $\times$ 2 sockets |
| **NVM** | 128 GB Optane DC Persistent Memory Module $\times$ 2 |
| **Optane SSD** | 480 GB Intel Optane SSD 900P |
| **NAND-flash SSD** | 3.2 TB Intel SSD DC P4610 |
| **DRAM** | 16 GB 2933 MHz DDR4 $\times$ per CPU socket |
| **Operating System** | Ubuntu 20.04 LTS with Linux kernel 5.1.0 |
| **Database System** | MySQL InnoDB 8.0.25 |

In addition, to observe how remote NUMA node access can affect the performance of DCPMMs, we intentionally use the `numactl` command to access the DCPMM using processors from other sockets, and run previous four FIO workloads.

Second, both direct I/O and DAX bypass the page cache. To investigate whether there is a performance difference between them, for DCPMM, we also test the performance of each file system under simple read/write workloads when turning on the `direct` option of FIO.

Third, we use filebench [15] as a macro benchmark to observe the performance of file systems working on DCPMM under more realistic workloads. We use four predefined workloads from filebench, namely, fileserver, varmail, webserver, and webproxy. In contrast to the simple workloads generated by FIO, there is a mix of reads and writes in the filebench workloads, and the request size varies between requests. The details of these workload configurations are listed in Table 2.

**Table 2.** Filebench workload configurations.

| | Fileserver | Varmail | Webserver | Webproxy |
|---|---|---|---|---|
| **# of files** | 500 K | 1 M | 500 K | 1 M |
| **meandirwidth** | 20 | 1 M | 20 | 1 M |
| **average file size** | 128 K | 32 K | 64 K | 32 K |
| **# of thread** | 16 | 16 | 16 | 16 |
| **R/W Ratio** | 1:2 | 1:1 | 10:1 | 5:1 |

- Fileserver emulates the I/O activities of a file server. It is a write-intensive workloads that mixed operations of create, write, read, delete, and append.
- Varmail represents mail server workload that saves each email in a separate file. The workload consists of create, delete, append, and fsync operations.
- Webserver is a read-intensive workload that consists of open, read, close, and log append activities.

- Webproxy represents the I/O activities of a simple web proxy server. The workload consists of create, write, open, delete, and log append operations.

Then, because only DCPMMs on the same socket can be organized as an interleaved region, to explore the performance of DCPMMs on different sockets when they work together, we use the Linux device mapper (i.e., dm-stripe) to combine DCPMMs located in different sockets into one logical device and run FIO to evaluate the performance.

Finally, we use the TPC-C benchmark [35] to evaluate how MySQL [36] works on DCPMMs with traditional and NVM-aware file systems. By default, we set the size of the buffer pool as 1 GB and the flushing method as direct I/O (O_DIRECT). We create 500 warehouses. The warm up time is set as 180 s, and the execution time is set as 10 min. We change the page size and consistency settings to observe the impact of these configurations on performance.

## 4. Evaluation

### 4.1. Micro-Benchmark Results

In this section, we use FIO [14] to generate simple workloads, such as sequential read/write and random read/write, to observe the best throughput of file systems on DCPMMs, Optane SSDs, and NAND-flash SSDs. Figure 1 shows the performance of file systems on DCPMMs, Optane SSDs, and NAND-flash SSDs.

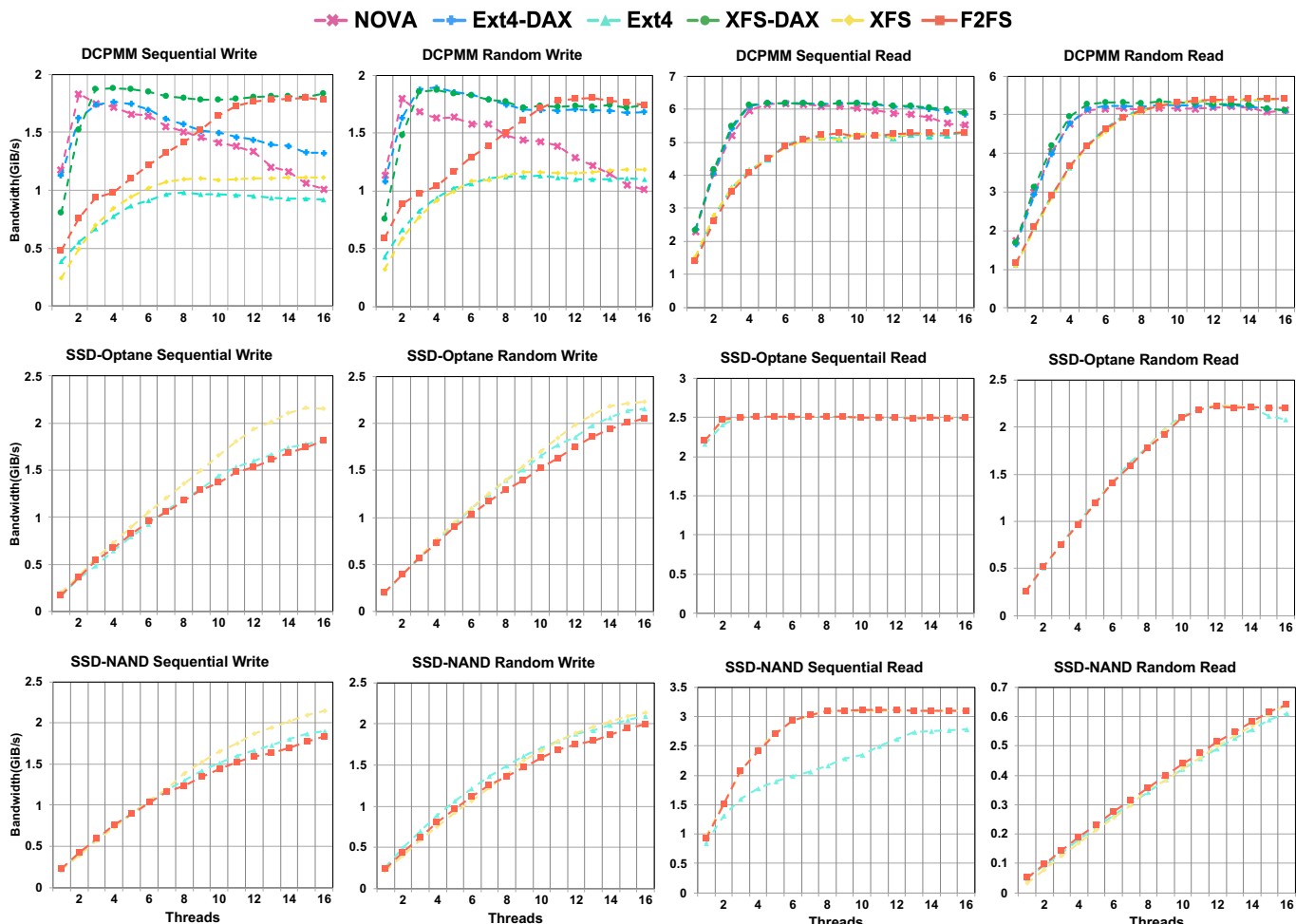

**Figure 1.** The read/write performance of traditional and NVM-aware file systems on Optane DCPMM, Optane SSD, and NAND-flash SSD.

First, we focus on the read and write performance of file systems on DCPMMs. From the datasheet of DCPMMs, the read and write bandwidth of the 128 GB Optane DCPMM are 1.85 GB/s and 6.8 GB/s, respectively [37]. As shown in the figure, read and write operations show different characteristics. In terms of write operations, there is a significant performance gap between traditional file systems and NVM-aware file systems. The best performance of Ext4 and XFS is only about 60% of that of their NVM-aware versions. In contrast, both traditional file systems and NVM-aware file systems show good read performance on DCPMMs. NVM-aware file systems outperform the traditional file system by up to 20% in the sequential read workload; with the random write workload, there is no significant different in the peak performance between the two types of file systems.

We infer that the performance difference between traditional file systems and NVM-aware file systems is mainly due to the page cache. Because traditional file systems use page cache, data are copied twice when I/O is performed, once from the user space to page cache and once from the page cache to DCPMMs, which introduces overhead. However, for read workloads, first, readahead can reduce access to storage devices by prefetching continuous data; second, subsequent reads of the same data can be handled by the page cache, which reduces latency and access to storage devices. Therefore, the performance gap for read workloads is smaller.

From the above evaluations, we obtain the following observation: *direct access achieves a significant improvement in file system write performance, but has a smaller impact on read performance.*

The number of threads accessing DCPMMs at the same time also affects the performance of file systems. In terms of write operations, the three NVM-aware file systems , Ext4-DAX, XFS-DAX, and NOVA, can utilize the full write bandwidth of DCPMMs when the number of threads is around 4. As the number of threads increases, XFS-DAX can maintain the full bandwidth but the performance of Ext4-DAX and NOVA drops dramatically. In terms of read operations, file systems can reach the best performance at 8 threads. After that, the performance is flat or slightly decreases as the number of threads increases, but there is no drastic performance drop as in the case of write operations.

We assume that the cause of this performance pattern is the contention in the integrated memory controller (iMC). As we introduced in Section 2.1, DCPMMs are installed in the memory slot and communicate with CPU through iMC. Requests to DCPMMs are queued in the read/write pending queues that are maintained by iMC. Because the iMC can only allow a limited number of simultaneous accesses to a channel, when the number of threads increases, the later threads need to wait for the preceding threads to complete their requests. Since the read performance of DCPMMs is significantly better than the write performance, requests in the read pending queue are processed faster, so read requests are less affected by simultaneous access than write requests.

From the above evaluations, we obtain the following observations:

- *A small number of threads is enough to saturate the write bandwidth of one DCPMM. In some situations, more threads may even degrade the performance.*
- *The read performance of DCPMMs is greatly superior to its write performance. Concurrent read access does not degrade the read performance significantly.*

Third, we compare how file systems work on the Optane SSDs and NAND-flash SSDs. Since these two devices are not NVM devices, we only test XFS, Ext4, and F2FS. The write performance of file systems on these two devices is similar, and they all rise as the number of threads rises. The read performance is much different. In terms of sequential read workload, both Optane SSDs and NAND-flash SSDs perform well because of the benefits of readahead. With fewer threads, file systems perform better on Optane SSD than on NAND-flash SSD. However, the performance of NAND-flash is better when the number of threads increases. In terms of random read workload, the performance of file systems on Optane SSD is significantly better than that on NAND-flash SSD, with a nearly four-times difference in optimal performance between them.

Because DCPMMs sit in the memory bus, access from remote NUMA nodes can affect the performance. Figure 2 shows the performance of file systems when using processors in a remote socket to access DCPMMs. As shown in the figure, remote access has a different performance impact on read and write operations. For read operations, there is almost no performance difference between remote access and local access. However, in terms of write operation, remote access significantly degrades performance, regardless of whether it is a traditional file system or an NVM-aware file system. The writing performance of remote access is only 60% of that of local access, and as the number of threads increases, the performance drops sharply.

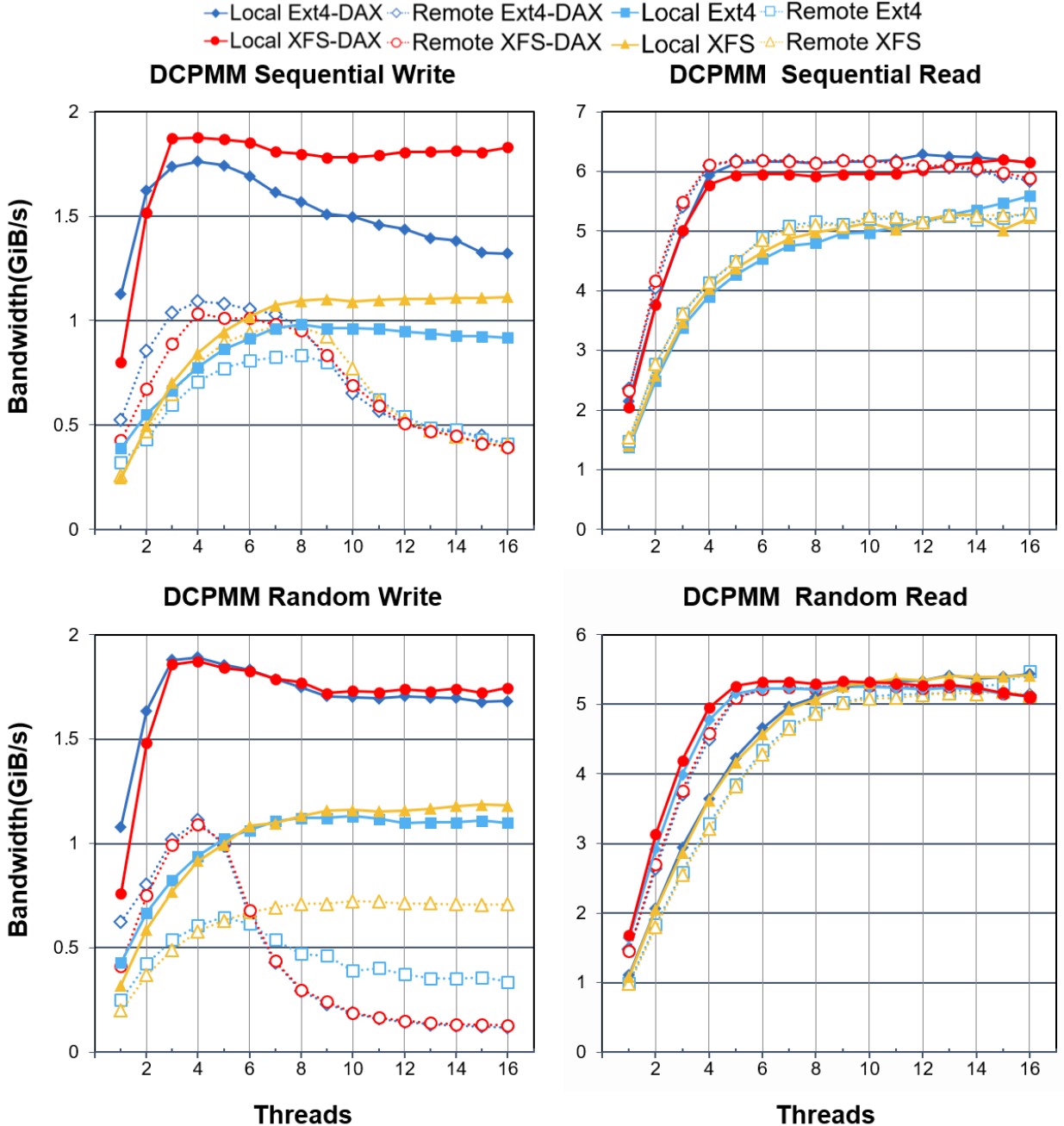

**Figure 2.** The performance of traditional and NVM-aware file systems on DCPMM when accessed from remote socket.

From the above evaluations, we get the following observation: *access from remote sockets reduces the write performance of DCPMMs but has little impact on the read performance.*

### 4.2. DAX vs. Direct I/O

Similar to DAX, direct I/O also bypasses the page. However, their call paths are not the same. Direct I/O avoids data copy between the page cache and user space, but it still goes through the generic block layer. Theoretically, this introduces an unnecessary overhead. To determine if this overhead has a performance impact, we evaluate the performance of each file system, using FIO with direct I/O.

Figure 3 shows the performance of file systems on DCPMMs when using direct I/O. As shown in the figure, when the page cache is bypassed, the performance of traditional file systems and the performance of their NVM-aware versions are similar to each other.

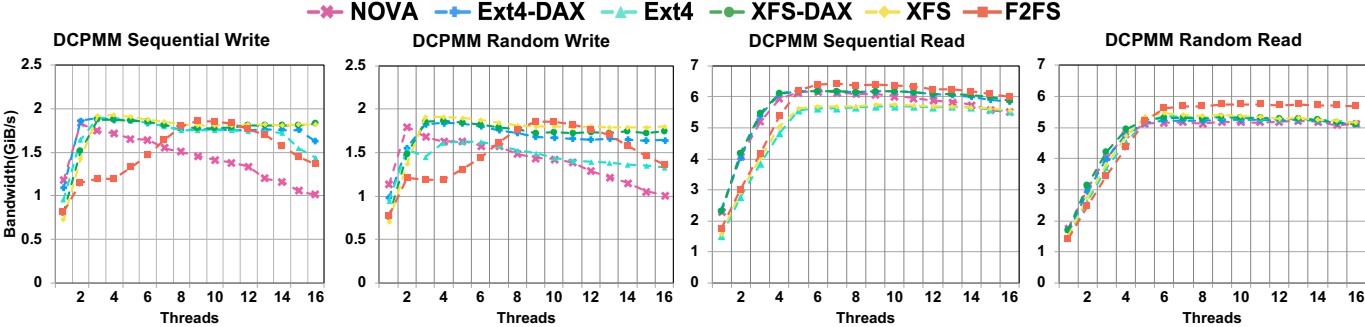

**Figure 3.** The read/write performance of traditional and NVM-aware file systems on Optane DCPMM when using direct I/O.

In terms of the two write workloads, XFS and XFS-DAX almost show the same performance. However, there is a performance gap between Ext4 and Ext4-DAX with the random write workload. We assume that the difference in performance between Ext4 and XFS comes from their different direct I/O implementations. In the kernel we are using (i.e., kernel 5.1.0), Ext4 uses `__generic_file_write_iter()` to send I/O request and then calls `generic_write_sync()` to make sure the previously completed write requests are on non-volatile storage before the next request starts. However, XFS uses `iomap_dio_rw()` to process I/O, which does not need `generic_write_sync()`.

For read workloads, only in the sequential read can we observe a consistent performance gap between traditional file system and their NVM-aware counterpart of about 5%. Since no one uses the page cache, we speculate that this small performance difference comes from the block layer and the alignment checking of direct I/O.

As a result, there can be a small performance gap between DAX and direct I/O, but it is dependent on the specific direct I/O implementation. We note that DAX has other advantages over direct I/O. First, the request size of direct I/O has an alignment restriction, so it is not available for all workloads. Second, DAX is byte-addressable, while direct I/O is not.

### 4.3. Macro-Benchmark Results

In order to observe how file systems on DCPMMs work in more realistic workloads, in this section, we generate requests by using filebench with fileserver, varmail, webserver, and webproxy workloads. The details of of each workload configuration are listed in Table 2.

Figure 4 shows the performance of each workload on different file systems by IOPS. As shown in the figure, NVM-aware file systems do not always outperform traditional file systems. For example, when running read-intensive workloads, such as webserver, all traditional file systems outperform NVM-aware file systems.

This is because read-intensive workloads benefit from page cache and readahead. Readahead enables read requests to prefetch the data adjacent to the current request into the page cache, thus it reduces the number of storage accesses, according to the spatial locality. Similarly, since the data are stored in the page cache, the subsequent accesses to

the same data are handled by the main memory, resulting in lower data access latencies. In contrast, because DAX bypasses the page cache, every I/O request has to access DCPMM.

From the above evaluations, we obtain the following observation: *DAX does not always provide better performance. Page cache may provide better performance for read-intensive workloads.*

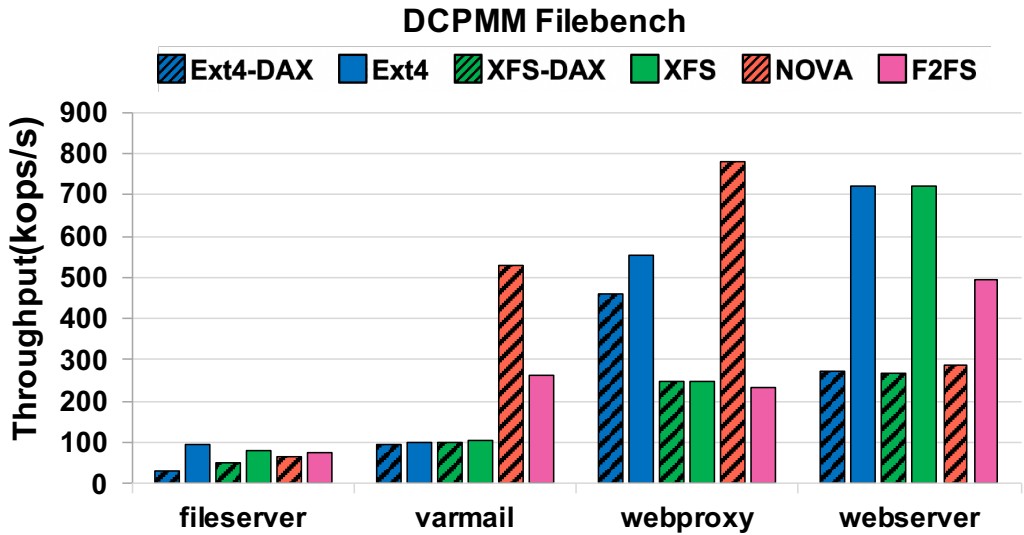

**Figure 4.** The filebench performance of file systems on DCPMMs.

Webproxy is also a read-intensive workload; however, NOVA outperforms traditional file systems that use page cache in this workload. We suspect the reason is that NOVA's metadata management strategy for NVM is efficient on workloads with a huge directory width, such as webproxy (i.e., 1 million files per directory). NOVA builds radix trees in DRAM for each directory inode; this strategy makes the file search operation very efficient, even for large directories. For the same reason, NOVA also has the best performance in varmail, a workload with large directories.

The IOPS of the fileserver workload is much lower compared with other workloads; however, because fileserver is a write-intensive workload, Ext4, XFS, F2FS, and NOVA have actually achieved the write bandwidth of DCPMMs. In workloads with high write ratios, such as fileserver and varmail, Ext4 and XFS have a similar or better performance than their DAX-enabled versions. We assume there are two reasons for this: first, traditional file systems can benefit from the page cache. Data from multiple requests are first buffered in the page cache and then written to the DCPMM in a single pass. Second, Ext4-DAX and XFS-DAX still use the journaling mechanism designed for block devices. In block-based journaling, a whole metadata block is stored into the journaling space in persistent storage, even if the metadata change affects only a single byte. This results in the updating of the metadata being accompanied by unnecessary data writes, resulting in write amplification [38].

From the above evaluations, we obtain the following observation: *file systems need to redesign their metadata management strategy to ensure data consistency while effectively utilizing the full potential of NVM devices.*

### 4.4. Remote Socket Stripping

To further increase the bandwidth of DCPMMs, Intel provides a technique called Interleaved Region. This technique can make multiple DCPMMs on a single socket appear as a single logical virtual address space. An interleaved region is similar to a RAID-0 device with a 4 KB stripe size. Because DCPMMs are managed by iMC, the limitation of interleave region is that only DCPMMs that sit in the same socket can be combined into a set. In order to use DCPMMs located in the different sockets as a group, one way is to use the Linux

device mapper to organize them as a single logical device. To date, there are two target drivers, dm-stripe and dm-linear, supporting DAX [39].

We configure two DCPMMs located in different sockets as a dm-stripe device. As with the interleaved region, the stripe size is set to 4 KB. We use FIO to evaluate the read and write performance of the dm-stripe device. The request size is fixed as 4 KB, and the number of threads varies from 1 to 32. We allow only the cores in the two sockets as the DCPMMs to access the logical device (16 cores per CPU, 32 cores total).

Figure 5 shows the read and write performance of the Ext4 file system on the dm-stripe DCPMM device. (We have followed the instructions given from Intel [39] to set up a DAX-aware dm-stripe device or LVM device. However, we could not enable the DAX feature successfully on the dm-stripe and LVM devices. We will evaluate the performance of DAX-enable file systems on the dm-stripe target device once we address the problem.)

Similar to the evaluation result in Section 4.1 when using a single DCPMM, there is no significant difference in performance between the sequential read and random read for the dm-stripe device. Performance increases as the number of threads rises, peaking when the number of threads reaches 12. The read performance can reach 11 GB/s; continuing to increase the number of threads does not affect the performance. The write performance improves and then decreases as the number of threads increases. The best write performance is achieved when the number of threads is 8, reaching 2.5 GB/s.

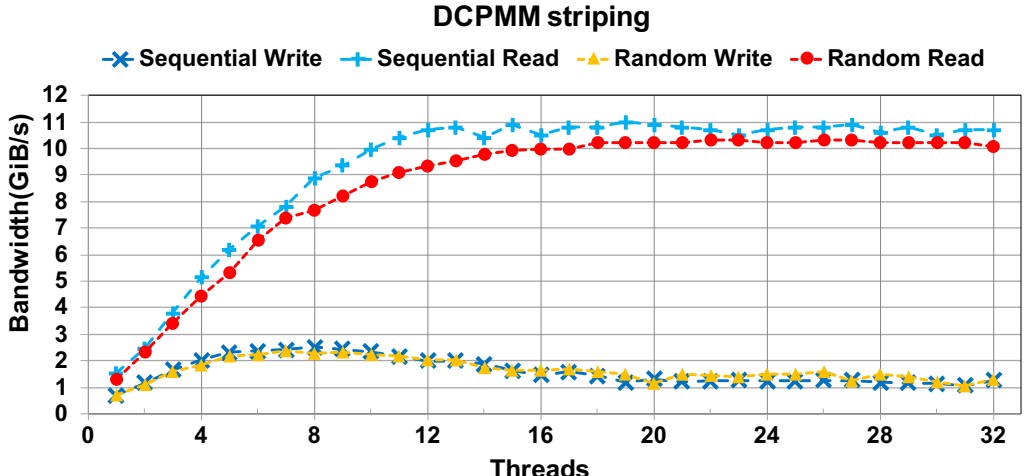

**Figure 5.** The read and write performance of the Ext4 file system on a dm-stripe DCPMM device that is organized by two DCPMMs sitting in different sockets.

As shown in Figure 1, when using one DCPMM, the best write performance of Ext4 is around 1.2 GB/s and the best read performance of Ext4 is around 5.3 GB/s. The dm-stripe device that combined with two DCPMMs almost doubles the performance of a single device.

From the evaluation, we obtain the following observation: *under proper settings, organizing the DCPMMs distributed in different sockets can also bring out the full performance of the devices.*

### 4.5. Database on DCPMMs

In this section, we evaluate the performance of MySQL [36] on DCPMMs with the Ext4 and Ext4-DAX file systems. To find the best settings to achieve the better database performance on DCPMMs, we analyze the performance of the database using the TPC-C benchmark with different configurations. The TPC-C workload [35,40] is an online transaction processing (OLTP) workload that involves a mix of five concurrent transactions. By default, we set the size of the buffer pool as 1 GB and the flushing method as direct I/O (O_DIRECT). We create 500 warehouses and set the warm-up time as 180 s and the

execution time as 10 min. We also use the same workloads and configurations on Optane SSDs and NVMe SSDs to compare with DCPMMs.

### 4.5.1. Effect of Page Sizes

A page is the basic unit of I/O performed by InnoDB. In MySQL InnoDB, users can choose different page sizes by changing the configuration. The default page size is 16 KB. Larger page sizes have better performance on HDDs because the main source of latency in disk is the seeking time of the disk head. However, as shown in Figure 6, storage devices such as DCPMM and SSDs that do not have mechanical components do not benefit from large page sizes. All three storage devices and both Ext4 and Ext4-DAX file systems reflect the same trend. The performance of the database system decreases as the page size increases. This is because large pages may access unnecessary data.

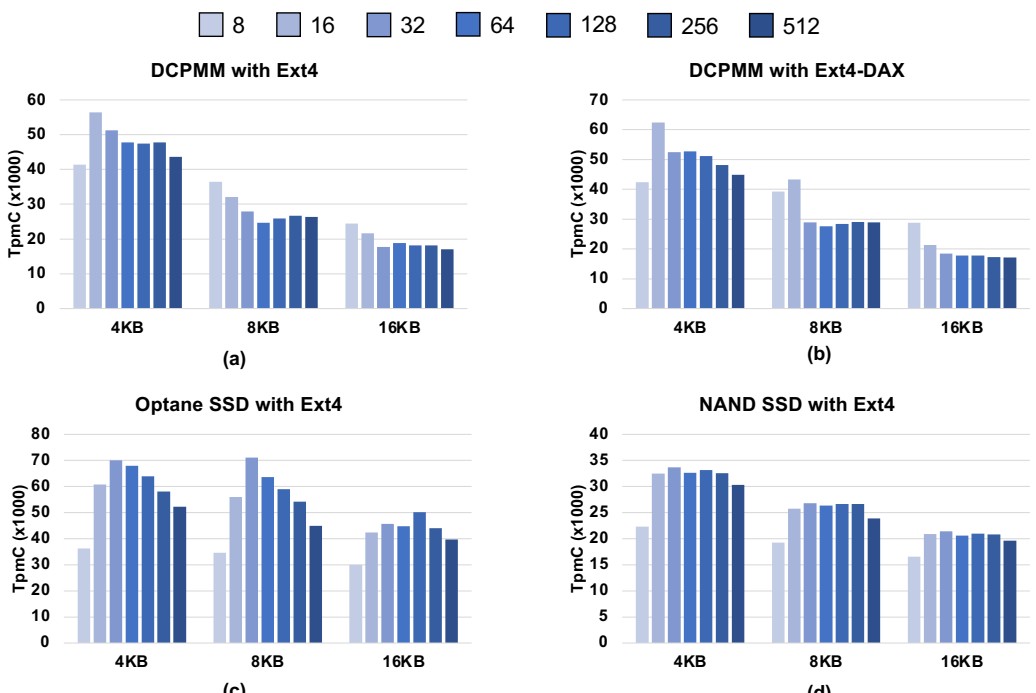

**Figure 6.** TPC-C results on DCPMMs (**a**,**b**), Optane SSDs (**c**), and NAND SSDs (**d**). The InnoDB page size varies from 4 KB to 16 KB. The number of connected clients varies from 8 to 512.

In particular, the performance of DCPMMs is very sensitive to the page size. As shown in Figure 6a,b, when using the Ext4 file system, the 4 KB page size has 76% and 160% transaction per minutes (tpmC) increases, compared to 8 KB and 16 KB page sizes in the case of 16 connected clients, respectively. When using the Ext4-DAX file system, the corresponding tpmC increase is 44% and 192%, respectively.

Compared to Optane SSDs and NAND SSDs, DCPMM is more sensitive to the page size, due to its ultra-low latency and inability to support too many simultaneous accesses. When the page size is small, data can be flushed into DCPMMs in a very short period of time, which in turn reduces memory controller contention to some extent. When the page size is larger, it takes longer to read and write a page, and may increase the number of page I/Os because of the possibility of accessing unnecessary data and the limited buffer space.

From the evaluation, we obtain the following observation: *DCPMM is better able to show its strengths from small size requests.*

### 4.5.2. Multiple Clients

We change the numbers of clients to see how it affects the performance of MySQL on DCPMMs. The number of clients varies from 8 to 512. As shown in Figure 6, DCPMMs and

Optane SSDs—the two devices that use 3D XPoint as the storage medium—and NAND SSDs reflect different patterns. As the number of clients increases, the tpmC of NAND SSDs first rises and then stabilizes. In contrast, the tpmC of DCPMMs and Optane SSDs rises and then falls. This result is in line with our evaluation in Section 4.1, where DCPMMs can achieve great performance with a small number of requests, but the performance decreases instead as the number of simultaneous requests increases.

As shown in Figure 6a,b, DCPMMs perform best when the page size is 4 KB and the number of clients is 16 (56,406 tpmC when using Ext4 and 62,378 tpmC when using Ext4-DAX). When the page size is 16 KB, the performance is best in the case of 8 clients. This confirms our previous observation that as the page size becomes larger, the time required for a single request becomes longer, and the contention for the memory controller increases.

### 4.5.3. Consistency Techniques

Ensuring data consistency is a top priority for databases. The MySQL InnoDB storage engine uses redundant writes to guarantee atomicity and utilizes frequent `fsync()` system calls to ensure durability. In this section, we evaluate and analyze how these mechanisms affect the performance of MySQL on DCPMMs, Optane SSDs, and NAND SSDs with Ext4 and Ext4-DAX file systems.

**Redundant writes for atomicity.** For consistency and recovery, the database should write pages to storage atomically. However, because a page is usually greater than the size of the atomic write unit that storage devices guarantee, if a system crash occurs during a page write, the page will have a mix of old and new data that are not known to the database. In this situation, the partial written page cannot be recovered by Write-Ahead Logging, thus the database cannot maintain data consistency. To resolve this problem, InnoDB provides a mechanism for writing a page atomically by redundant writes, which is called Doublewrite Buffer (DWB). The doublewrite buffer is a storage area where the database writes pages from the buffer pool before writing the pages to their proper positions in the data files. Although data are written twice, the doublewrite buffer does not require twice as much I/O overhead. Data are written to the doublewrite buffer in a large sequential chunk, with a single `fsync()` call to the operating system.

During the evaluation, we fix the number of clients to 16. Figure 7 shows the impact of the doublewrite buffer on the performance of DCPMMs, Optane SSDs, and NAND SSDs. NAND SSDs appear to receive very little impact from DWB, while DCPMM is heavily impacted by DWB. As shown in Figure 7a,b, relative to no DWB, when using the Ext4 file system, DWB reduces performance by 14.5%, 49.6%, and 36.8% for 4 KB, 8 KB, and 16 KB pages, respectively. When using the Ext4-DAX file system, DWB reduces performance by 9%, 45.8%, and 46.1% in the case of 4 KB, 8 KB, and 16 KB pages, respectively. When the page size is 4 KB, DWB has the least impact on performance. This coincides with the previous performance impact of the page size.

**Data flushing for durability.** The databases issue a `fsync()` system call after writing pages to persistent storage. The `fsync()` is expensive, but it is essential to databases, as it guarantees durability. InnoDB provides several data flushing methods to control how data are persisted to storage devices, which can affect the I/O throughput. O_DIRECT is the default method that bypasses page cache and issues `fsync()` every time after writing pages. Like O_DIRECT, the O_DIRECT_NO_FSYNC method also bypasses page cache, but skips the `fsync()` system call after each write operation.

It is reasonable to assume that using O_DIRECT_NO_FSYNC will improve performance, compared to using O_DIRECT. However, as shown in Figure 7c,d, using O_DIRECT_NO_FSYNC result in little performance improvement in our evaluation. There are two reasons for this result. First, MySQL uses a group commit technique that allows multiple simultaneous transactions to `fsync()` the log file once for all the transactions waiting for the `fsync()`, which reduces the number of calls to `fsync()`. Second, because removing all the `fsync()` is not suitable for file systems, such as XFS and Ext4, which require an `fsync()` system call to synchronize file system metadata changes, MySQL does some modifications to

the O_DIRECT_NO_FSYNC method after version 8.0.14. Even if O_DIRECT_NO_FSYNC is used, `fsync()` will still be called after creating a new file, after increasing the file size, and after closing a file. Hence, this method does not completely avoid `fsync()`.

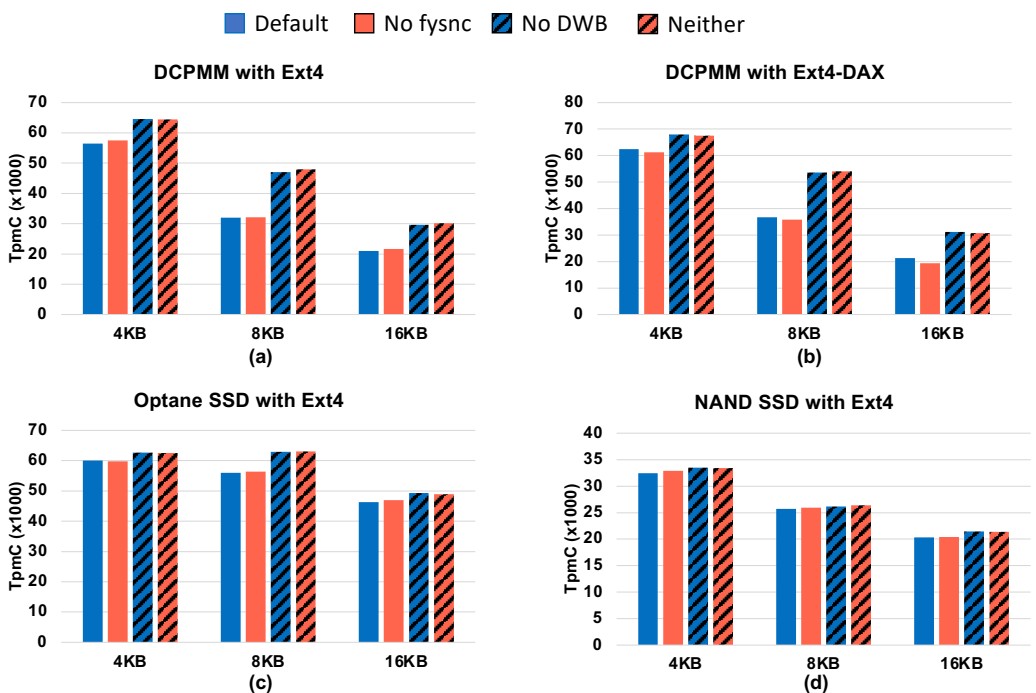

**Figure 7.** TPC-C results on DCPMMs (**a**,**b**), Optane SSDs (**c**), and NAND SSDs (**d**) with different consistency settings. The number of clients is fixed to 16. (*Default*: flush method is O_DIRECT and enable double write buffer; *No fsync*: flush method is O_DIRECT_NO_FSYNC and enable double write buffer; *No DWB*: flush method is O_DIRECT and disable double write buffer; *Neither*: flush method is O_DIRECT_NO_FSYNC and disable double write buffer.)

In Figure 7b, we can observe an interesting, but counter-intuitive, phenomenon. That is, on DCPMM with Ext4-DAX file system, the performance is lightly better with `fsync()` than without it. We assume that this is because the underlying implementation of `fsync()` is different. In non-DAX mode, data flows through the file system and storage stack as normal. In DAX mode, because the DCPMM is mapped directly to process's address space, data is exchanged directly between the CPU cache and the DCPMM. In order to provide the original semantics of the POSIX APIs while ensuring proper operation on DCPMMs, operating system developers have reimplement `fsync()` in DAX mode. More specifically, in Linux, the kernel uses the radix tree to hold DAX exceptional entries that track the state of the persistent-memory pages used by DAX. These exceptional entires store several pieces of information, such as the page size, the sector offset within the persistent-memory storage, and some flags. From these exceptional entires, DAX knows which dirty pages need to be flushed from the processor cache when an `fsync()` is called from user space, which avoid unnecessary page flush. In addition, DAX `fsync()` uses `CLWB` instruction to flush out the cache line. On some platforms, `CLWB` will retain the flushed out cache line in the cache hierarchy in non-modified state, which reduces the possibility of cache miss on a subsequent access. For the above reasons, using `fsync()` in DAX mode may instead bring a slight performance boost. However, it is important to note that the performance improvement depends on the specific API implementation and hardware platform.

From the above evaluations, we get the following observation: *Even with the same APIs, we need to consider their different impact on normal block devices and NVM devices under different hardware platforms.*

**Database atomicity and durability on DCPMMs.** When using databases on DCPMMs, doublewrite buffer and `fsync()` are required to guarantee data consistency. Even

though the ADR feature of Intel platforms can flush the write-protected data buffers and place the DRAM in self-refresh, it is limited in the amount of data it can flush and, more importantly, it does not save CPU caches. If there is no doublewrite buffer, it is possible that pages are mixing old and new data. There is also a risk of having data only written to the processor caches without the cache flush command that guarantees data durability.

Figure 7a,b show that disabling `fsync()` after each write does not bring noticeable performance gains. Doublewrite has a significant impact on DCPMMs and is related to the page size. Consequently, we recommend using database on DCPMMs with the DWB and O_DIRECT options on and choosing a smaller page size to balance the trade-off between data atomicity, data durability, and system performance.

### 4.5.4. Effect of Direct Access

As shown in Figure 6a,b, the best performance of DCPMM with Ext4-DAX outperforms that of DCPMM with Ext4 by 10.6%, 18.7%, 17.8% when using 4 KB, 8 KB, and 16 KB pages, respectively. DAX removes the extra copy operation by performing reads and writes directly to the storage device, which operates like memory to get the lowest latency. Thus, DCPMM in DAX mode has better performance, but also more sensitive to settings that affect latency. For example, when the page size is 8 KB, database gets the best performance at 16 clients, instead of 8 clients in non-DAX mode. However, when the optimal performance is reached, as the number of clients increases, there is a huge slump in performance when DAX mode is used. In contrast, the performance drop in non-DAX mode is much smoother. Specifically, the drop between the optimal and suboptimal performance is 9.1%, 12.1%, 11.2% for 4 KB, 8 KB, and 16 KB page sizes in Ext4, while the difference is 16.1%, 33.2%, 25.9% for Ext4-DAX, respectively.

## 5. Conclusions

In this article, we evaluate the performance of commonly used traditional file systems and NVM-aware file systems on Intel Optane DCPMMs. First, we use micro benchmark to exploit the performance of file systems on DCPMMs with simple read/write workloads. In addition, we explore how NUMA affects the performance of DCPMMs. Second, we use filebench as a macro benchmark to evaluate performance of file systems on DCPMMs with more realistic workloads. Third, we investigate the performance a logical device formed by DCPMMs located on different socket, which we call remote socket stripping. Last, we evaluate the performance of MySQL InnoDB on DCPMMs with traditional file systems and NVM-aware file systems. In addition, we compare the performance characteristics of DCPMMs with those of Intel Optane SSDs and NAND-flash SSDs. From these evaluations, we obtain the following observations that may be helpful for future system designs:

- From the micro benchmark evaluation, we observed the following:
  1. Direct access has a significant improvement in the file system write performance but has a smaller impact on the read performance.
  2. A small number of threads is enough to saturate the write bandwidth of one DCPMM. In some situations, more threads may even degrade the performance.
  3. The read performance of DCPMMs is greatly superior to its write performance. Concurrent read access does not degrade the DCPMM read performance significantly.
  4. Accessing from remote sockets reduces the write performance of DCPMMs significantly, but has a small impact on the read performance.
- From the macro benchmark evaluation, we observed the following:
  5. Page cache may provide better performance than DAX for read-intensive workloads.
  6. A low overhead metadata management strategy is needed to fully utilize the potential of NVM devices.
- From the remote evaluation of socket stripping, we observed the following:

> 7. Under proper settings, organizing the DCPMMs distributed in different sockets can also bring out the full performance of the devices.

- From the evaluation of databases on DCPMMs, we observed the following:

> 8. DCPMM is better able to show its strengths from small size requests.
> 9. Even with the same APIs, we need to consider their different impact on normal block devices and NVM devices under different hardware platforms.

**Author Contributions:** Conceptualization, G.Z. and Y.S.; methodology, Y.S. and G.Z; investigation, G.Z. and S.L.; validation, J.H.; visualization, S.L. and G.Z.; writing—original draft preparation, G.Z.; writing—review and editing, G.Z. and Y.S. All authors have read and agreed to the published version of the manuscript.

**Funding:** This work was supported in part by the National Research Foundation of Korea (NRF) Grant funded by the Korean Government (MSIT) (No. 2018R1C1B5085640, 2021R1C1C1010861). This research was supported through BK21 FOUR (Fostering Outstanding Universities for Research) Program funded by Ministry of Education of Korea (No. I20SS7609062) (corresponding author: Yongseok Son).

**Conflicts of Interest:** The authors declare no conflict of interest.

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
