# Peer review of "An Empirical Evaluation of NVM-Aware File Systems on Intel Optane DC Persistent Memory Modules†"

_electronics, doi:10.3390/electronics10161977_

Round 1

Reviewer 1 Report

The paper is an evaluation of various storage technologies focusing on two types of Optane and SSDs.  Overall, the results compare and contrast for a single node the performance of each for file-system type operations.  The work provides some clear conclusions that could help others in deciding what technologies are best for their use and ways to improve the Optane technology by leveraging better SW.

The paper is very sound in its methodology, but the presentation could use work.  It somewhat reads like a data dump though there is some analysis.  In addition, graphs often show one technology at a time and it would be good to have some graphs that compare multiple hardware technologies.  A bit more of helping the reader out to see the points more clearly would go a long way to improving the paper.

While not needed for this paper a look at how DAOS and other emerging technologies and parallel file systems can use this would be interesting.  Are the issues the same or different?

Author Response

Thank you for your valuable comments.

Please see the attachment for response details.

Reviewer 2 Report

The research, as is presented, is focused primarily on presenting and interpreting the results; but it describes just briefly the way of generating reads and writes; moreover, what “complex workload” was considered during test is also not sufficiently explained, motivated, and discussed. Please detail these aspects more specifically in the paper.

The conclusions of the paper are somehow unstructured; since the tests in the research paper were organized into four sections: simple read/write operations, complex workload, remote socket stripping and database performance, it would have been better to organize conclusions with respect to each aspect that was tested during the experiments. Using this approach, the results of each category of tests would have been better summarized.

I suggest moving the Related work chapter form the paper at the beginning of the paper; maybe, if possible, it could be merged or included in the background chapter.

Author Response

(The authors gave the same response as above.)

Reviewer 3 Report

The paper presents a good analysis of the performance of filesystems on Intel's 3D XPoint media in NVDIMM form-factor. Overall, this is of only moderate interest to the community as this combination of media and form-factor is mainly intended to be used as persistent memory and not as storage behind a filesystem. Nevertheless, the analysis does corroborate the ineffectiveness of the combination by showing that performance of filesystems is as good as or better with a 3D XPoint SSD form-factor.

With regards to the analysis, there are also a few shortcomings that should ideally be addressed:

  • The configuration is too small: only 1 DCPMM DIMM per socket. Given the importance of parallelism and interleaving in DIMM configurations, it would be more representative to evaluate configurations with more than one DCPMM DIMM per socket.
  • There is little point in comparing DAX and non-DAX filesystems on read workloads if you bypass the page-cache (page 4). What exactly is the objective of this comparison? How can you then conclude (page 6) that direct access does not improve reads, if all your experiments are without the page-cache?
  • What is the meaning of "remote socket" experiments with Optane SSD in Figure 2? The SSD does not sit on any socket, but on the PCIe bus.
  • What exactly is happening in the experiments of Section 4.3? What is the point in striping across sockets if you only allow each CPU to access DCPMM on its own socket? What sort of striping is taking place then? Even if CPUs can stripe across sockets, don't NUMA effects offset any gains from striping?

Author Response

(The authors gave the same response as above.)

Reviewer 4 Report

In the paper, authors concentrate on comparing the performance characteristics between traditional block-based file systems and NVM-aware file systems on DCPMMs. In addition, authors also compare the DCPMMs with Intel Optane SSDs and NAND-flash SSDs. Finally, authors summarize seven observations from the evaluation results and performance analysis. Overall, the paper is interesting and has done some work.

Author Response

Thank you for your valuable comments so that we can improve our paper.

Round 2

Reviewer 3 Report

The revisions to the experiments and text address my concerns to a satisfactory level. A more thorough version would contain experiments with multiple DCPMMs and would experiments with highly-tuned NVRAM-aware filesystems such as DAOS, but the paper is acceptable as is.

Author Response

Thank you for your valuable comments.

Because we do not have enough DRAMs in our experiment machine, we can only use the configuration that has one DCPMM per socket.
In our paper, we evaluate the performance of multiple DCPMMs by organizing two DCPMMs in different sockets as a logical device, as shown in Figure 5 (DCPMM Stripping) in Section 4.4.
We will evaluate the performance under different configurations that have multiple DCPMMs in one socket in future work after we have purchased the new components.

In this paper, all the file systems that we evaluate are local file systems. Considering the evaluation of DAOS which is a distributed storage system, we think it can be another research topic. Thus, in our future work, we will investigate how DAOS and parallel file systems exploit DCPMM effectively.